# Fabrication and Characterization of EVA Resins as Adhesives in Plywood

**DOI:** 10.3390/polym15081834

**Published:** 2023-04-10

**Authors:** Yu Zhang, Ye He, Jiayan Yu, Yuxin Lu, Xinhao Zhang, Lu Fang

**Affiliations:** 1College of Furnishings and Industrial Design, Nanjing Forestry University, Nanjing 210037, China; 2Co-Innovation Center of Efficient Processing and Utilization of Forest Resources, Nanjing Forestry University, Nanjing 210037, China

**Keywords:** wood–plastic plywood, hot-press, secondary press, thermoplastic resin, physical–mechanical properties, aldehyde-free

## Abstract

The practical problem of free formaldehyde pollution in the plywood industry is that polyethylene films have been shown to be able to replace some urea–formaldehyde resins for wood adhesives. To broaden the variety of thermoplastic plywood, reduce the hot-press temperature, and save energy consumption, an ethylene–vinyl acetate (EVA) film was selected as a wood adhesive to manufacture a novel wood–plastic composite plywood via hot-press and secondary press processes. The effects of the hot-press and secondary press processes at different levels on the physical–mechanical properties of EVA plywood (tensile shear strength, 24 h water absorption, and immersion peel performance) were evaluated. The results showed that the properties of the resulting plywood using the EVA film as an adhesive could meet the type III plywood standard. The optimum hot-press time was 1 min/mm, the hot-press temperature was 110–120 °C, the hot-press pressure was 1 MPa, the dosage film was 163 g/m^2^, the secondary press time was 5 min, the secondary press pressure was 0.5 MPa, and the secondary press temperature was 25 °C. EVA plywood can be used in indoor environments.

## 1. Introduction

With the rapid development of the construction and furniture industries, the demand for wood has intensely increased. Wood-based panels could effectively improve the broad utilization rate of wood. Wood-based panels mainly include plywood, particle board, and fiberboard. Among them, plywood is widely used in the furniture, construction, packaging, car, and boat manufacturing industries owing to its excellent physical and mechanical properties. However, plywood production was mainly formaldehyde-based-material-type adhesives, of which urea–formaldehyde (UF) resin adhesives account for 80% of the total use [1]. UF resins are characterized by low cost, mature technology, and a great gluing effect, but they release free formaldehyde during their production and use. To reduce the formaldehyde emission from wood-based composites, several researchers have investigated the optimization process of aldehydes in adhesives through the reduction of the formaldehyde–urea molar ratio [2], the control of the reaction temperature and pH [3], and the addition of a formaldehyde trapping agent [4,5] of UF resins during synthesis. These can effectively reduce the emission of free formaldehyde from wood-based panels.

With the improved living conditions of people, most countries have issued more stringent environmental protection standards. The formaldehyde emission standard for composite wood products issued by the United States in 2017 stipulates that the formaldehyde emission of hardwood plywood manufactured with single or composite cores should be ≤0.05 ppm. According to the GB/T 39600-2021 classification of formaldehyde emissions from wood-based panels and their products implemented in China, the formaldehyde content has been newly classified, and the highest electric network frequency level limits the formaldehyde emission from exceeding 0.025 mg/m^3^. Under the advocacy of energy conservation and emission reduction and green environmental protection policies, green and environment-friendly adhesives have received considerable attention [6,7,8]. Presently, isocyanate (MDI) adhesives [9], soybean protein adhesives [10,11], starch adhesives [11,12,13], and other biological adhesives [14] have been widely used in the production of wood-based panels, and related products have been marketed. In addition, inorganic adhesives, such as silicate, magnesium oxychloride, and phosphate, have received attention from scientific researchers owing to their mildew resistance, water resistance, and flame retardancy. Zheng et al. [15] prepared bamboo chips/magnesium oxychloride composites with higher mechanical properties and water resistance using magnesium oxychloride gel, bamboo chips, and 0.3% polycarboxylate superplasticizer, providing research guidance for new wall materials.

In recent years, thermoplastic resin films, such as polyethylene (PE), polypropylene (PP), and polyvinyl chloride (PVC), have been widely used for producing plywood owing to their excellent water resistance, flexibility, easy processing, and secondary melting characteristics. PE has the simplest structure and has been widely studied as a wood adhesive [16,17,18,19]. Fang et al. [17] systematically evaluated the adhesive properties of high-density polyethylene films. The results showed that the plywood manufactured with PE as an adhesive featured similar bonding strength and elastic modulus with the UF resin plywood containing similar resin contents. In addition, because its plasticity can endow the plywood with stronger resistance to bending damage, PE thermoplastic plywood exhibits a higher modulus of rupture. PVC is a thermoplastic resin characterized by high flame retardancy and chemical resistance and can be used for wood veneer bonding. Gao [20] obtained the optimal process conditions for the production of PVC–thermoplastic plywood via a response surface method. The result showed that the properties of the produced plywood can meet the type II plywood standard, and the optimum hot-press conditions were 170 °C and 1 min/mm. Compared with PE and PVC films, the PP film features higher heat resistance. The bonding strength of the PP thermoplastic plywood is 1.5 MPa after three treatment cycles (immersing in boiling water for 4 h, then drying at 63 °C for 20 h and immersing again in boiling water for 4 h), which can meet the requirements of ordinary plywood category I in the GB/T 9846-2015 standard [21]. However, owing to the high melting temperature of the PP film, the temperature of the PP thermoplastic plywood should not be lower than 180 °C. In the manufacturing process of thermoplastic resin plywood, the heat transfer rate and uniformity can directly affect the plywood performance. Li et al. [22] not only increased the mechanical interlocking between the plastic and wood but also significantly reduced the hot-press time of the plywood through the perforation of the PVC film and then bonding with the wood veneer. Ye et al. [23] used mechanical methods to punch holes in the wood veneer surface and then combined it with the PE film. The increase in holes on the veneer was highly conducive to the penetration of PE, increasing the number of dendritic glue nails, which can form a highly stable microstructure, and thus, the panel strength is improved. Conventional wood–plastic composites can be obtained by extrusion molding or injection molding [24,25]; however, thermoplastic plywood is usually prepared by hot-press and secondary press processes. Bekhta et al. [26] used a high-density polyethylene film as an adhesive to manufacture alder plywoods, investigated the effects of hot-press temperatures and hot-press times on the physical and mechanical properties of alder plywood panels, and compared these properties with UF resin and phenol–formaldehyde resin plywoods. In a study on a hot-press process factor of a wood–plastic composite plywood, Chang [27] selected the secondary press conditions as follows: a secondary press time of 5 min, a secondary press pressure of 1 MPa, and a secondary press temperature of 30 °C; Fang [28] used a PE film to manufacture a poplar plywood and selected a hot-press pressure and secondary press pressure of 1 MPa. Among them, the selection of the secondary press conditions has a certain impact on the plywood performance, but studies on the secondary press process are few at present.

The main bonding mechanism between wood veneer and thermoplastic film is mechanical interlock, no chemical reaction. When a specific thermoplastic film is selected as a plywood adhesive, a suitable hot-press temperature needs to be selected so that the thermoplastic resin can flow into the wood pores of the wood to form glue nails that give the plywood mechanical strength. Usually, the hot-press temperature should be higher than the melting point of the thermoplastic film. Fang et al. [17] chose a hot-press temperature of 160 °C for the preparation of PE plywood. At this time, the PE film has better mobility and forms a tight bond with the wood veneer. Compared with the PE film, the melting temperature of the PP film was higher. Song et al. [29] prepared PP plywood at a hot-press temperature of 165–195 °C based on the melting temperature of the PP film. At 185 °C, it was found to have higher tensile strength than those formed at 165 °C due to the lower viscosity and deeper penetration of PP at 185 °C. Poly-β-hydroxybutyrate film (PHBF) is a biodegradable thermoplastic that can also be used as an adhesive for aldehyde-free plywood. Chen et al. [30] prepared veneer–PHBF composite properties under a hot-press temperature at 170 °C. Results showed that the properties of the produced plywood can meet the type II plywood standard. The mechanical properties and water resistance of these thermoplastic plywoods are excellent, but the processing temperature is usually above 160 °C, which not only consumes more energy to process, but also causes surface discoloration of the plywood. Ethylene–vinyl acetate copolymer (EVA) is a thermoplastic resin with a relatively low melting temperature and remarkable flexibility. In this study, an ethylene–vinyl acetate copolymer (EVA) film was selected as an adhesive instead of the traditional formaldehyde adhesive to prepare the EVA wood–plastic plywood. The effects of the hot-press and secondary press processes at different levels on the physical–mechanical properties of the EVA plywood were evaluated. The purpose of this study was to provide a new type of thermoplastic plywood with excellent performance, simple preparation, and no formaldehyde environmental protection, and to provide a theoretical basis for production practice.

## 2. Materials and Methods

### 2.1. Materials

Poplar veneers were purchased from Minsheng Wood Industry Co., Ltd. (Shandong, China). The dimensions were 300 × 300 × 1.6 mm^3^, and the moisture content was 6–8%. The EVA film purchased from Huakai Supply Chain Co., Ltd. (Shenzhen, China), had a thickness of 0.1 mm and a density of 0.91 g/cm^3^.

### 2.2. Production of the EVA Wood–Plastic Plywood

A three-layer wood-based plywood was assembled by three poplar veneers and EVA films. The plywood was prepared via a combination of hot-press and secondary press methods (Figure 1). The hot-press pressure was controlled to 1 MPa, and the hot-press temperatures were 90, 100, 110, 120, 130, and 140 °C. The numbers of EVA films were 1 layer, 2 layers, and 3 layers (one layer of EVA film was equivalent to 81.5 g/m^2^ of double-sided sizing) during the hot-press process. Then, the plywood was secondary-pressed at room temperature for 5 min under 1 MPa.

The best secondary press time for a pretest was 5 min. Under the same hot-press conditions, the EVA wood–plastic plywood was prepared under secondary press temperatures of 25, 45, 65, and 85 °C and cold pressures of 0.5, 1, 1.5, and 2 MPa.

### 2.3. Characterization

#### 2.3.1. Thermal Properties of the EVA Film

Melting temperature: A differential scanning calorimetry (DSC) analysis of the EVA film (5–10 mg) was conducted under a nitrogen atmosphere using DSC-250 (TA Instruments, New Castle, DE, USA). The EVA film was heated up from 0 to 180 °C at a rapid heating rate of 10 °C/min and preserved at 180 °C for 5 min to remove the thermal history of the sample. Then, the EVA film was cooled down from 180 to 0 °C at a rate of 10 °C/min and preserved at 0 °C for 5 min. Finally, it was heated up again from 0 to 180 °C at a rapid heating rate of 10 °C/min to obtain the melting point of the EVA film.

Heat stability: A thermogravimetric analysis of the EVA film was conducted using TGA-250 (TA Instrument) under a nitrogen atmosphere at a heating rate of 10 °C/min from 25 to 800 °C. Then, 5–10 mg of the EVA film was used for the test.

#### 2.3.2. Physical–Mechanical Characterization

The physical–mechanical properties of the EVA plywood (tensile shear strength, wood failure ratio, 24 h WA, and immersion peel performance) were evaluated according to the Chinese National Standard (GB/T 17657-2013) [31]. Before the test was conducted, all specimens were conditioned at 20 °C and 65% relative humidity for 48 h.
(1)Mechanical strength: According to the requirements of GB/T 9846-2015 [32] “Ordinary Plywood” type II and III plywood standard, the tests for plywood were performed under the conditions of 63 ± 3 °C hot water immersion for 3 h and 20 ± 3 °C cold water immersion for 24 h. The result revealed that the plywood could not meet the type II bonding strength test.(2)Twenty-four-hour water absorption: The size of three-layer plywood specimens with dimensions of 100 mm × 100 mm was weighed in 20 °C water before and after 24 h of soaking mass m_1_ and m_2_. The 24 h water absorption is calculated using the following equation:(1)$$\mathrm{WA}\left(\%\right)=\frac{m_{2\;}-m_{1}}{m_{1}}\times 100\%$$(3)Immersion peel performance: The immersion peel performance of the EVA wood–plastic plywood was tested according to the Chinese National Standard GB/T 9846-2015 [32], and the peeling delamination between the adhesive layers of the specimen was observed.

#### 2.3.3. Scanning Electron Microscopy (SEM)

Two layers of plywood with parallel structures were prepared as SEM observation samples under all process conditions. The interface structure of the plywood was examined using a Quanta-200 ESEM (Hillsboro, OR, USA). The specimens were fixed onto the copper sheet with adhesive tape and sprayed with gold.

## 3. Results and Discussion

### 3.1. Characterization of the EVA Film

The thermal properties of the EVA film had a vital impact on the preparation and performance of the EVA wood–plastic plywood. The preparation process of the EVA thermoplastic plywood indicated that EVA films melted and softened at a higher temperature and pressure to penetrate the wood veneer, which endows the EVA wood–plastic plywood with mechanical strength. Figure 2a shows the melting curve of the EVA film and its peak temperature (melting temperature) of 84.8 °C. Based on this, hot-press temperatures of 90, 100, 110, 120, 130, and 140 °C were set.

EVA consists of ethylene and vinyl acetate monomers (VA) in the presence of initiators for high-pressure polymerization, where the VA content affects the material properties and crystallinity [33]. Studies have shown that as the VA content was between 1% and 40%, EVA featured high transparency, high flexibility, and high viscosity, which is used in packaging films, hot-melt adhesives, agricultural land films, and coatings [34,35]. From the EVA pyrolysis curve, the thermal decomposition of EVA occurred in three stages (Figure 2b). The first stage was at 0–280 °C, and the mass loss rate of EVA was within 1%. The second stage occurred between 300 and 380 °C, at a temperature of 380 °C, and the weight loss rate of EVA was 15.6%; moreover, EVA ester bond breakage released an acetic acid and generated a mixture of CO_2_ and CH_4_ [36]. The VA content of the EVA film used in this study was 22.3% based on the weight loss rate of the acetic acid. In the final stage, PE began to decompose at 420–550 °C and ended as the temperature reached 600 °C, indicating that EVA has a wide processing temperature range.

A previous study noted that the thermal decomposition of most wood and other natural fibers is between 215 and 310 °C [37]. In order to avoid pyrolysis of plywood, the hot-press temperature should be chosen below 215 °C.

### 3.2. Effect of the Hot-Press Process on the Performance of the EVA Wood–Plastic Plywood

#### 3.2.1. Effect of the Hot-Press Process on the Bonding Strength of the EVA Wood–Plastic Plywood

##### Hot-Press Temperature

In order for the thermoplastic resin to fully flow into and between the wood pores and form a glue nail structure, the hot-press temperature should be 15 to 35 °C higher than the melting temperature of the thermoplastic. Therefore, in this paper, the hot-pressing temperature for EVA wood–plastic plywood ranged from 90 to 140 °C.

The hot-press temperature showed a low effect on the dry tensile shear strength of the EVA wood–plastic plywood (Figure 3). With increasing hot-press temperatures from 90 to 110 °C, the dry strength of the EVA plywood increased from 1.09 to 1.2 MPa with a 11.7% variation, and the hot-press temperature continuously increased, with no significant change in dry strength. Because EVA had melted and entered the porous structure of the plywood, it formed a mechanical interlock structure, which endows the plywood with dry strength.

The main bonding mechanism between wood veneer and EVA film was mechanical interlock, not chemically bonded [29,38,39,40], and its bonding interface has poor resistance to water molecule damage, resulting in the wet strength of the plywood being significantly lower than the dry strength. The hot-press temperature had an excellent influence on the stability of the glued structure (Figure 3a). At a very low hot-press temperature (90 °C), the stability of the bonding structure of the wood was poor, and the bonding strength could not meet the type III standard of plywood in GB/T 9846-2015. The wood failure ratio of the EVA wood–plastic plywood prepared under this condition was almost 0 (Figure 3b). There was a large gap between the bonding interfaces of plywood (Figure 4a,b). At hot-press temperatures between 110 and 120 °C, the bonding strength of type III plywood was 0.9 MPa, reaching a stable state, and its wood failure ratio significantly increased, indicating that the fluidity and permeability of EVA films were enhanced at higher hot-press temperatures. The mechanical interlock structure formed by the wood veneer was more stable, and the gap between the bonding interface was smaller (Figure 4c,d). Because the EVA melt viscosity was high, the continuous increase in the hot-press temperature on its permeability improvement was not significant (Figure 4e,f). With increasing hot-press temperatures from 110 to 140 °C, the improvement rate of wet strength of the plywood was <10%. Song et al. [29] prepared wood veneer/PP film composites using wood veneer and PP film. When the hot-pressing temperature was 20 °C higher than the melting temperature of the PP film, the tensile shear strength of the wood veneer/PP film composites was the best. This conclusion is consistent with the conclusions of this study.

##### Dosage of the EVA Film

EVA film was used as an adhesive in the EVA wood–plastic plywood, and its dosage had a significant effect on the number of mechanical nails and the thickness of the wood interface layer. The plywood with different film dosages was prepared under a hot-press temperature of 120 °C, hot-press time of 1 min/mm, and hot pressure of 1 MPa, and its bonding strength is shown in Figure 5. With increasing dosages from 81.5 to 244.5 g/m^2^, the dry strength of the plywood increased from 1.19 to 1.24 MPa, with a 4% increase. Therefore, the EVA film that could endow the bonding strength of the panel was limited. With increasing EVA dosages, the number of mechanical nails increased, but the change in bonding strength of the wood was small. An EVA film dosage of 81.5 g/m^2^ could meet the requirements of the bonding interface of the mechanical interlock structure.

The hydrophobicity of the EVA film blocked the entry of water molecules and resisted the damage of water molecules to the bonding interface. All samples exhibited strength higher than 0.7 MPa for wet tests, which met the requirement for type III–grade plywood of GB/T 9846-2015 standard (Figure 5a). With increasing EVA dosages from 81.5 to 163 g/m^2^, the wet strength of the wood increased from 0.82 to 0.91 MPa. However, with an increasing film dosage of 244.5 g/m^2^, the wood adhesive strength decreased by 7%; because of the high viscosity of EVA, its fully molten state was attained after a longer time, and the shorter hot-press time led to an incomplete molten state. Moreover, too thick a layer of glue would weaken the adhesion between veneers. Therefore, with increasing EVA film dosages, its bonding strength decreases. The effect of the dosage of the thermoplastic film used on the bonding properties of another thermoplastic plywood is similar. Fang et al. [41] laminated the silane-modified poplar veneer with PE film, and when the PE thin film increased from 1 to 4 layers, the adhesive layer was more likely to detach from the modified poplar veneer, causing a decrease in bonding strength.

##### Hot-Press Time

The EVA thermoplastic plywood was prepared with different hot-press times under the conditions of a hot-press temperature of 110 °C, hot-press pressure of 1 MPa, and EVA film dosage of 163 g/m^2^. The hot-press time had a significant impact on the bonding strength of the panel (Figure 5b). The dry and wet strength of the EVA plywood first increased and then decreased with the hot-press time. With increasing hot-press times from 0.5 to 1 min/mm, the dry and wet strength of the plywood increased by 14% and 22%, respectively. This phenomenon is elucidated as follows: At the same hot-press temperature, the heat transfer from the surface layer of the slab to the core layer occurred at a certain amount of time, the short hot-press time resulted in a low slab temperature of the core layer, and incomplete molten EVA and the veneer could not form a mechanical interlock structure. With increasing hot-press times, the heat was completely transferred to the core layer, and the permeability of the EVA film in the wood pores became stronger, resulting in the thinning of the panel glued interface layer, which reduces the wood bonding strength. Therefore, with an increasing hot-press time of 2 min/mm, the dry and wet strength (type III) of the plywood decreased. The result showed that the hot-press time affects the penetration depth of the thermoplastic film. Overpenetration may occur if the hot-press time is too long, which will have a negative impact on the bonding strength of the thermoplastic plywood [22].

#### 3.2.2. Effect of the Hot-Press Process on the Water Absorption of the EVA Wood–Plastic Plywood

The water absorption could evaluate the dimensional stability of the plywood, which has a vital impact on the long-term use of the plywood. With increasing EVA film dosages, hot-press temperature, and hot-press time, the water absorption of the plywood gradually decreased, and the water resistance increased. Among them, the water absorption of the plywood was mainly affected by the amount of EVA film. With increasing EVA film dosages from 81.5 to 244.5 g/m^2^, the 24 h water absorption decreased by 23%, because water-repellent materials did not absorb water (Figure 6). The water absorption mainly occurred in the EVA wood–plastic plywood. As the EVA film was combined with the poplar veneer, one part of the EVA film fully penetrated the micropores of the wood veneer, and the other part covered the wood surface; therefore, the contact area between the wood and water molecules became smaller, which can reduce the absorption rate of water molecules in the wood. With increasing EVA film contents, the effect of blocked water molecules increased, and thus, the water absorption rate of the panel became lower. This conclusion was confirmed in the paper by another researcher [40], who found that plywood density also has an effect on water absorption: the water absorption of plywood panels decreased with increasing density.

The hot-press time and the hot-press temperature had a slight effect on the water absorption of the panel as the amount of the EVA film and wood components was constant. The increase in these conditions increased the penetration depth of the EVA film, and the interfacial compatibility of the board was also improved owing to the increase in the hydrophobicity, the volatilization of a small number of extracts in the wood, and the reduction of hydrophilic hydroxyl groups under a high temperature. However, the water absorption of the wood was still dominant; therefore, with increasing hot-press temperatures from 90 to 140 °C, the 24 h water absorption decreased by 6%, and with increasing hot-press times from 0.5 to 2 min/mm, the 24 h water absorption decreased by 3%.

#### 3.2.3. Effect of the Hot-Press Process on the Immersion Peel Performance of the EVA Wood–Plastic Plywood

The immersion peel performance was a vital indicator for evaluating the water resistance and gluing properties of the plywood. The immersion peel test (type Ⅲ) performed on all EVA plywood samples showed no evidence of delamination and degumming, which still met the lowest requirement of the GB/T 9846-2015 standard (the total length of each side of each specimen peeled from the same adhesive layer should not exceed 25 mm). Although the glue layer could resist the damage of water molecules, some water molecules in the glue layer could not destroy the board glue interface, which preserved its glue interface.

The type II impregnation peel test (soaking at 63 ± 3 °C for 3 h and then drying at 63 ± 3 °C for 3 h) produced stresses that caused the plywood to peel to varying degrees (Figure 7). Among them, the plywood manufactured at 90 °C could not meet the standard requirement of type II plywood. This phenomenon is elucidated as follows: At a lower hot-press temperature, the adhesive layer of the bonding strength of specimens was less than the wet expansion and dry shrinkage stress, and the panel layer produced peels. With increasing hot-press temperatures, the veneer and EVA film closely combined, which can partly resist stress. The panel with different EVA film dosages was peeled at the ends, and the peeling length was approximately the same, indicating that the amount of the EVA film has little impact on the immersion peel performance of the plywood (Figure 8). However, the EVA film was not completely melted at a low hot-press time, resulting in cracking and delamination on the panel. At hot-press times of 0.5 min/mm and 1 min/mm, the immersion peel strength of the panel was 22 and 14.4 mm (Figure 9). With increasing hot-press times from 1.5 to 2 min/mm, a little crack was observed on the panel.

### 3.3. Effect of the Secondary Press Process on the Performance of the EVA Wood–Plastic Plywood

The temperature of the wood panel decreased after the EVA thermoplastic plywood was subjected to a hot-press process. The thermoplastic film shrinks during the formation of the adhesive interface, and the shrinkage residual stress affects the stability of the adhesive interface [42]. A suitable secondary press process could reduce the shrinkage stress of the panel bonding interface and ensure the mechanical strength of the plywood. Therefore, accurate control of the secondary press process is a key factor for maintaining the stability of the mechanical mesh structure of the interface layer and the mechanical properties of the panel. A literature search showed that most researchers have systematically studied the hot-press process of thermoplastic plywood [17,22,29,43], but there is no in-depth discussion on the secondary press process. Based on the hot-press process, the effect of the secondary press process on the tensile shear strength of EVA wood–plastic plywood was studied. In this study, the optimum conditions for preparing the EVA thermoplastic plywood were a hot-press temperature of 110 °C, hot-press pressure of 1 MPa, hot-press time of 1 min/mm, EVA film dosage of 163 g/m^2^, and secondary press time of 5 min, varying with the secondary press temperature and secondary press pressure.

At low temperatures, the crystalline curing of the meshing structure at the glued interface of the sheet was complete. At a lower secondary press temperature, the dry and wet strength of the EVA thermoplastic plywood was relatively large, because the EVA film was completely crystallized at 25 °C, and the glued interface of the adhesive nail structure maintained a stable state (Figure 10a). With increasing cold press temperatures, the EVA film at a certain cold press time was not completely crystallized, but a part of the crystallization process was complete under no pressure conditions; moreover, the interface layer of the EVA film was damaged by shrinkage stress (Figure 11). At a cold press temperature of 85 °C, close to the melting point of EVA, the increased hot-press times without cold press could not eliminate the contraction stress of the glued interface of the EVA film.

A suitable secondary press pressure could resist the shrinkage stress of the panel bonding interface and ensure the mechanical strength of the plywood. Figure 10b shows the influence of the secondary press pressure on the bonding strength of the panel. The bonding strength of the secondary pressed panels was higher than that of nonsecondary pressed panels; particularly, a significant difference was observed between the bonding strengths of the secondary pressed panels (0.5 MPa) and nonsecondary pressed panels. With increasing secondary press pressures from 0.5 to 2 MPa, the dry and wet bonding strength of the panel decreased, because the greater the secondary press pressure, the faster the plate cooling rate, resulting in a highly intense shrinkage of the EVA film, and the interface layer gaps partly caused the glue nail failure; moreover, the panel from the hot-press to the secondary press processes without pressure caused the EVA film to rebound. With excessive secondary press pressure, the molten EVA film from the glue interface overflow formed a thin glue layer, which reduces the bonding strength of the panel.

## 4. Conclusions

EVA film can be used as an ideal wood adhesive to produce formaldehyde-free wood–plastic plywood. The hot-press and secondary press conditions have a significant impact on the physical and mechanical properties of EVA wood–plastic plywood. The hot-press temperature and hot-press time have an interactive effect on the EVA wood–plastic plywood tensile shear strength. With a hot-press temperature of 110–120 °C and a hot-press time of 1 min/mm, EVA melted fully and did not overpenetrate, and poplar veneer formed a good mechanical interlock structure. With the increased EVA film dosage, the number of bonded joints formed also increased. With a film dosage of 163 g/m^2^, the plywood bonding strength was optimal. Moreover, to reduce the shrinkage stress of the panel interface layer, a secondary press temperature of ~25 °C and a secondary press pressure of 0.5 MPa should be selected. According to the GB/T 9846-2015 standard, the tensile shear strength of the plywood should meet the type III plywood standard. EVA wood–plastic plywood can be used as a green panel for an indoor environment. Compared with thermoplastic plywood, such as PE, PP, and PVC plywood, the hot-press temperature of EVA plywood is 40 to 60 °C lower than other thermoplastic plywoods, which is more energy saving and consumption reducing. However, it also led to EVA plywood samples showing evidence of delamination and degumming after immersion in hot water at 63 °C for 3 h. In order to improve the mechanical properties of EVA plywood, chemical modification or using EVA/PE blends as plywood adhesives can be used to improve the interfacial adhesion of plywood in a subsequent work.

## Figures and Tables

**Figure 1 polymers-15-01834-f001:**
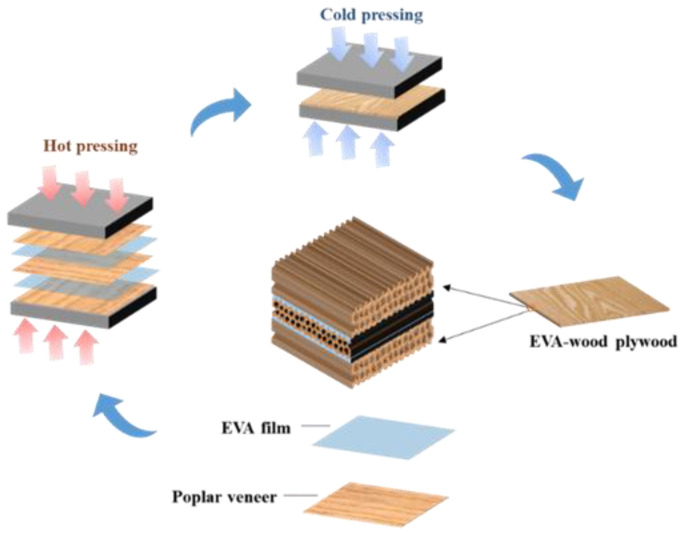
The process of plywood preparation.

**Figure 2 polymers-15-01834-f002:**
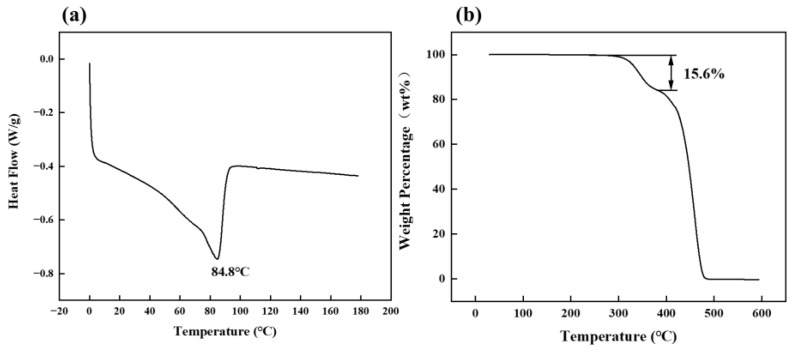
(**a**) Differential scanning calorimetry; (**b**) thermogravimetric curves of EVA film.

**Figure 3 polymers-15-01834-f003:**
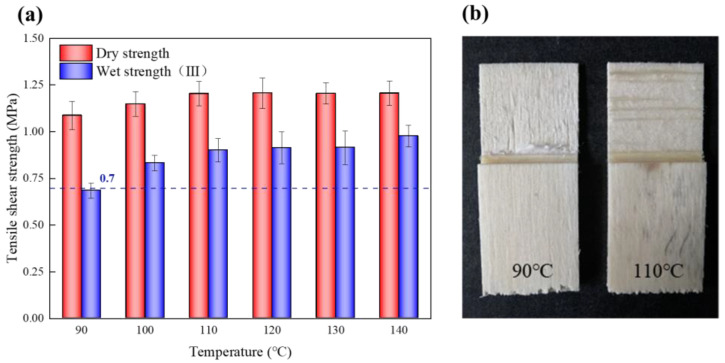
(**a**) Tensile shear strength of plywood at different hot-press temperatures; (**b**) wood failure rate of EVA–wood plywood (III).

**Figure 4 polymers-15-01834-f004:**
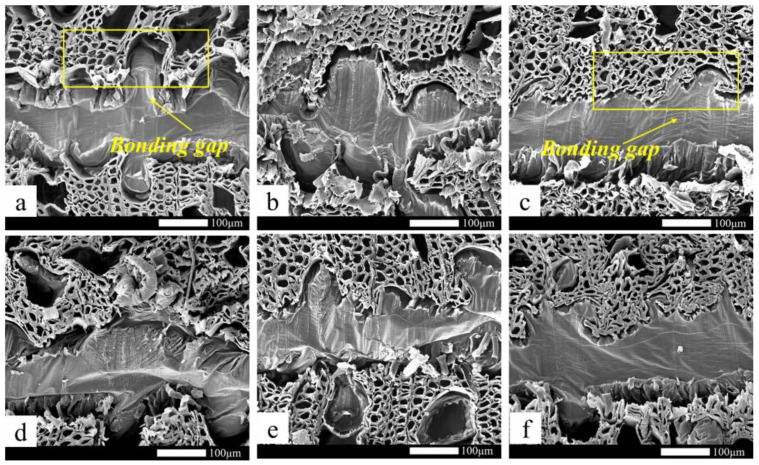
The interfacial surface of plywood with different hot-press temperatures: (**a**) 90 °C, (**b**) 100 °C, (**c**) 110 °C, (**d**) 120 °C, (**e**) 130 °C, (**f**) 140 °C.

**Figure 5 polymers-15-01834-f005:**
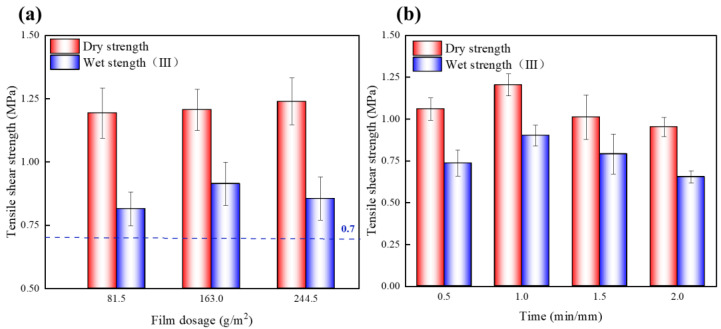
(**a**) Tensile shear strength of plywood at different film dosages; (**b**) tensile shear strength of plywood at different hot-press times.

**Figure 6 polymers-15-01834-f006:**
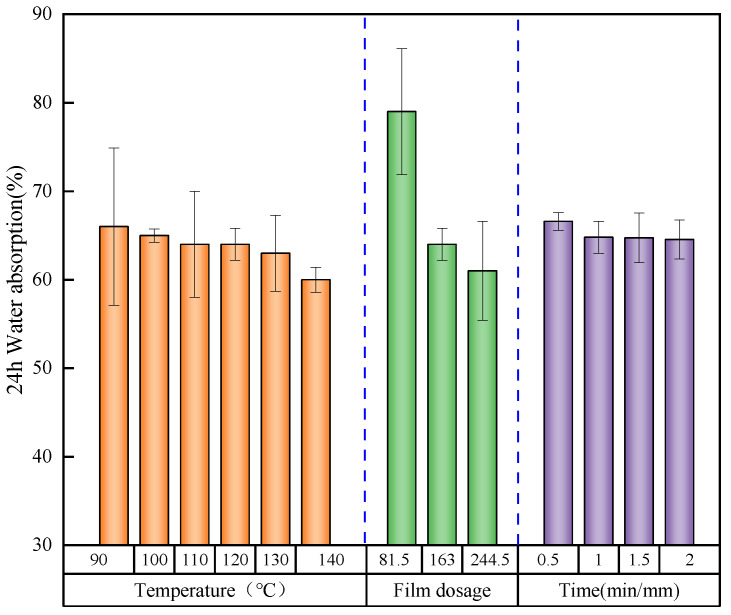
24 h water absorption of plywood under different conditions.

**Figure 7 polymers-15-01834-f007:**
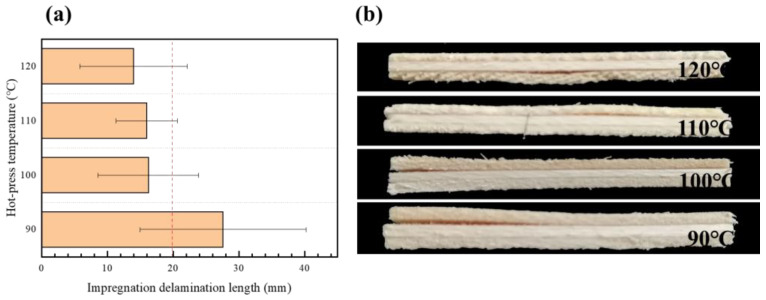
(**a**) Effect of hot-pressing temperature on impregnation peel performance; (**b**) impregnation delamination length under different hot-pressing temperature conditions.

**Figure 8 polymers-15-01834-f008:**
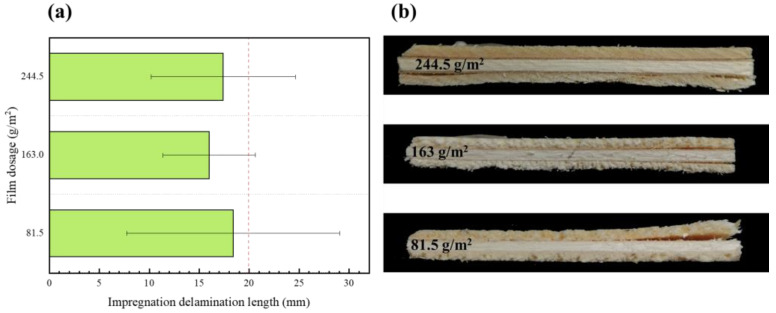
(**a**) Effect of EVA film dosage on impregnation peel performance; (**b**) impregnation delamination length under different film dosage conditions.

**Figure 9 polymers-15-01834-f009:**
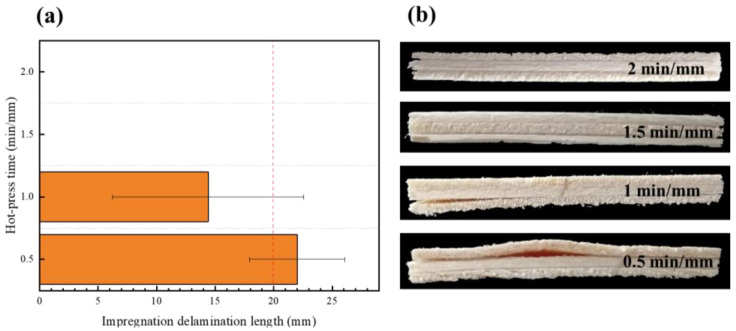
(**a**) Effect of hot-press time on impregnation peel performance; (**b**) impregnation delamination length under different hot-press times.

**Figure 10 polymers-15-01834-f010:**
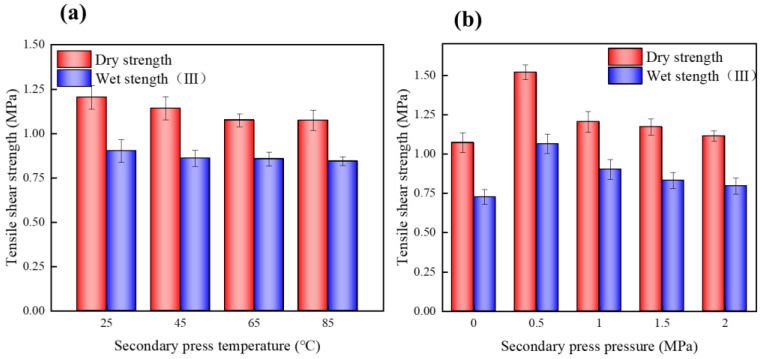
(**a**) Effect of secondary press temperature on tensile shear strength; (**b**) effect of secondary press pressure on tensile shear strength.

**Figure 11 polymers-15-01834-f011:**
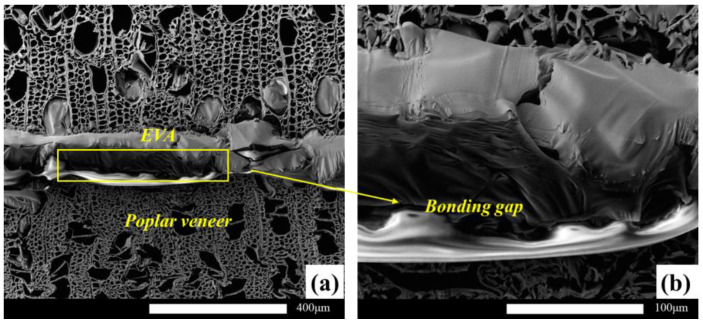
The bonding interface of nonsecondary press plywood. (**a**) ×150; (**b**) ×600.

## Data Availability

Not applicable.

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
