# Peer review of "Fabrication and Characterization of EVA Resins as Adhesives in Plywood"

_polymers, 2023, doi:10.3390/polym15081834_

Round 1

Reviewer 1 Report

Dear Authors,

I send you comments and  suggestions as noticed below:

Line 29 – remove „minus„

Lines 30-101 – missed the DOI number and links on all PhD dissertations;

Line 125 – give a short explanation „Why is the best cold-press time includes 5 min?“;

Line 131 – put the space after 2.3.1 „Thermal properties of ….“;

Line 141 - put the space after „comma “„5-10 mg ….“;

Line 145, 147-148, 157 - I suggest putting Chinese National Standards (testing and materials) in Reference List;

Line 158 – put tab after line 158;

Line 161 – the last part of sentences should be improved “… the process conditions in 2.2.”;

Line 174 – explain abbreviation in Fig. 2 “DSC” and “TG” of melting EVA film curve;

Line 175-179 – please put Reference for this statement EVA consists of ethylene and vinyl acetate monomers (VA) in the presence of initiators for a high-pressure polymerization, where the VA content affects the material properties and crystallinity. Studies have shown that as the VA content was between 1% and 40%, EVA featured high transparency, high flexibility, and high viscosity, which is used in packaging films, hot melt adhesives, agricultural land films, and coatings.”

Line 184 – missed the DOI number for 32nd Reference;

Line 188-190 – put tag after subtitle 3.2 and after subtitle 3.2.1;

Line 194-195 – this part of sentences isn’t understandable “…. with a 59 11.7% variation, ….”;

Line 242 – put tag before subtitle 3.2.1.3;

Line 258 – only if you can mark type III of plywood standard with dry strength measurements or put (III) in the middle of dry/wet strength;

Line 290-290 – I suggest putting afteron “…all 290 EVA plywood samples…” (type II & type III);

Line 309 – mean values of impregnation delamination length (mm) in Figure 7 are not aligned with the statements inside paragraph “…., the immersion peel strength of the panel was 22 and 14.4 mm.”

Line 340 - in Figure 9a, put on the axis ordinate the same keywords “Tensile shear strength (MPa)”

Line 344 – in Figure 10 you must mark two different pictures, left as “a” and right as “b” like in Figure 4.;

Line 386-453 - need to be improvement with articles DOI numbers and open access links od PhD dissertations.

The conclusion has to be improved, the environment-friendly significance of EVA adhesive has to be mentioned and the authors need to suggest further experiments and point out the  new ideas for executing a research projects.

Reviewer 2 Report

General comment:

The authors of this paper describe the effect of effects of the hot-press and cold-press processes at different levels on physical-mechanical properties of the EVA plywood. I confirm that paper has merit but needs a revision.

Each paragraph is not amply described and not correctly discussed. I invite authors to strengthen their sections.

Title: It is correctly clear and brief.

Abstract: Abstract requires a minor revision to introduce better the aim of the study and  the comprehension of the work. – Please use apex 163 g/m2

Keyword: Words from the title should not be used as keywords.

Introduction

The Introduction should provide a complete state of art what is being discussed in this paper. In this format, the introduction section is not complete. Some sentences are obvious and not necessary; other sentences can be useful for discussion section moving from here.

 The processing of raw materials …………….. rate of wood. Not clear………………this is a comment…………

Where is explained the aim of the study? Please, add more information and objective of the study.

Need rephrasing the introduction paragraph to show the aim of the paper and to emphasize the specific context.

M&M - Results

Why poplar was used?

300 × 300 × 1.6 mm3 volume or dimension?

What is the characteristic of hot-press ? name, model, kWh,….etc…….

The thermal properties of the EVA film had a vital impact on the preparation and performance of the EVA wood-plastic plywood. What means this comment? It is appropriate in the result paragraph?

Section Discussion is not developed; in Conclusion paragraph several sentences repeat general concept. The conclusions should be written comparing objects of the study and the results obtained. The conclusion is quite short basically including the results. More future developments and conclusions should be considered.

Round 2

Reviewer 2 Report

The paper had been significantly improved by the authors - congratulations to you.

This remains an important research topic and I appreciate the work of the authors.

The authors did improve the suggested review significantly.

Author Response

非常感谢您对本文的评论,这大大提高了论文的质量。